# PETformer: Long-term Time Series Forecasting via Placeholder-enhanced Transformer

## Abstract

Recently, the superiority of Transformer for long-term time series forecasting (LTSF) tasks has been challenged, particularly since recent work has shown that simple models can outperform numerous Transformer-based approaches. This suggests that a notable gap remains in fully leveraging the potential of Transformer in LTSF tasks. Consequently, this study investigates key issues when applying Transformer to LTSF, encompassing aspects of temporal continuity, information density, and multi-channel relationships. We introduce the Placeholder-enhanced Technique (PET) to enhance the computational efficiency and predictive accuracy of Transformer in LTSF tasks. Furthermore, we delve into the impact of larger patch strategies and channel interaction strategies on Transformer's performance, specifically Long Sub-sequence Division (LSD) and Multi-channel Separation and Interaction (MSI). These strategies collectively constitute a novel model termed PETformer. Extensive experiments have demonstrated that PETformer achieves state-of-the-art performance on eight commonly used public datasets for LTSF, surpassing all existing models. The insights and enhancement methodologies presented in this paper serve as valuable reference points and sources of inspiration for future research endeavors. Anonymous source code repository is available at: https://anonymous.4open.science/r/PETformer-main-4BF5.

## 1 Introduction

Long-term time series forecasting (LTSF) highlights a wide range of applications across diverse fields such as transportation, weather, energy, and healthcare, and has remained a prominent topic in academic research for an extended period (Zhou et al., 2021). Owing to advancements in deep learning, Transformer-based models have achieved groundbreaking results in various deep learning fields (Vaswani et al., 2017; Lin et al., 2022), and this trend has also extended to the LTSF domain (Wu et al., 2021; Zhou et al., 2022; Woo et al., 2022; Du et al., 2023).

Despite attempts to encourage the use of Transformer in LTSF, their efficacy has been challenged by Dlinear (Zeng et al., 2023), which attained unexpected results through a single-layer feedforward neural network, surpassing the state-of-the-art Transformer-based models of its time. Subsequently, PatchTST (Nie et al., 2023) inherited Dlinear's channel-independent technique and introduced the concept of patching from computer vision (Dosovitskiy et al., 2020), significantly enhancing Transformer performance in LTSF tasks and reaffirming the viability of Transformer. While the achievements of PatchTST are encouraging, there remains a lack of comprehensive understanding within the community regarding the optimal utilization of Transformer in the LTSF domain. To this end, this paper endeavors to investigate the following aspects and offer appropriate solutions.

**Temporal continuity**  LTSF tasks exhibit consistent and continuous temporal dependencies between input and output data. The original Transformer encoder-decoder architecture was initially developed for natural language processing tasks, utilizing positional encoding and decoder recursion to maintain temporal relationships in the input and output sequences, correspondingly (Vaswani et al., 2017). While prior dominant Transformer-based LTSF models follow this architecture (Zhou et al., 2021; Wu et al., 2021; Zhou et al., 2022), recent studies indicate that the cross-attention mechanism between the encoder and decoder restricts the performance of LTSF (Li et al., 2023).

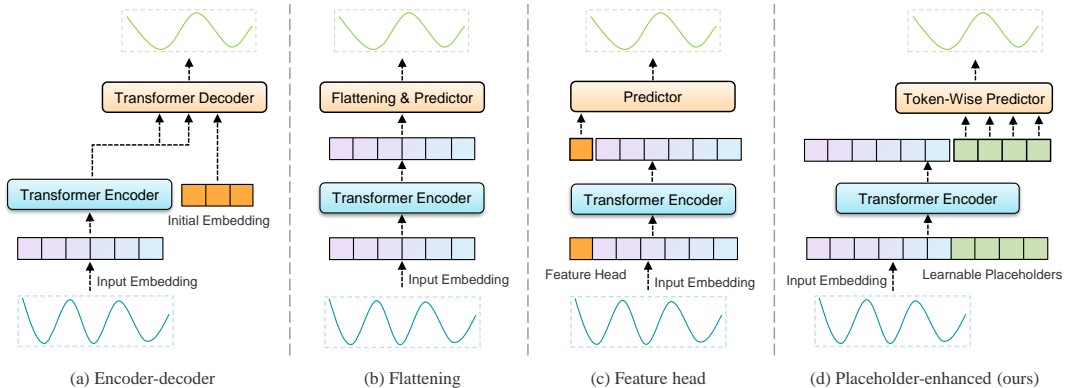

(a) Encoder-decoder     (b) Flattening     (c) Feature head     (d) Placeholder-enhanced (ours)

Figure 1: Comparison of different application techniques for Transformer in LTSF.

PatchTST addressed this concern by the Flattening technique, avoiding encoder-decoder information loss through the encoder-only architecture. However, the utilization of Flattening on lengthy sequences introduces a heavy linear-flatten-layer, resulting in an excessive increase in parameter count compared to the Transformer feature extraction module itself. To address this, the Feature Head[1] architecture emerges as a viable solution, preserving the advantages of an encoder-only architecture while mitigating the drawbacks of Flattening. Nonetheless, Feature Head struggles to fully preserve temporal features due to the compression of extensive temporal information into a single constrained vector, particularly for longer-term future horizons.

To overcome this limitation, we propose the Placeholder-enhanced Technique (PET), integrating both historical and future data segments as inputs to the Transformer model (see Figure 1). Each placeholder represents a segment of future data to be predicted. PET architecture, as opposed to other architectures, delivers advantages such as (i) improved computational efficiency, particularly through the token-wise predictor, which significantly reduces parameter requirements in the prediction head (by over 99% compared to Flattening architecture), and (ii) the functional fusion of Encoder and Decoder components, facilitating the direct modeling of relationships between historical and future data on a coherent temporal plane.

**Information density and multi-channel relationships**   Time series data, similar to images, is a low-density information source, as individual data points do not provide meaningful semantic information. PatchTST draws inspiration from the ViT approach in computer vision (Dosovitskiy et al., 2020), partitioning the original sequence into patches to extract sufficient semantic information. However, unlike static RGB channels in images, multivariate time series data (MTS) exhibits a variable number of channels with intricate relationships. Therefore, in contrast to ViT's direct channel mixing (DCM) approach, PatchTST inherits Dlinear's channel-independent approach to simplify the modeling of complex relationships among multiple variables.

In this work, we build upon PatchTST's patch and channel-independent strategies, with distinctions that (i) we explore larger patches, referred to as Long Sub-sequence Division (LSD), demonstrating improved performance through richer semantic information (ii) we investigate channel relationship modeling, termed Multi-channel Separation and Interaction (MSI), revealing the complexity of modeling relationships among multiple variables in time series data.

In summary, this paper makes the following contributions:

- From the perspectives of temporal continuity, information density, and multi-channel relationships, we systematically explore the optimal application architecture for Transformer in the context of LTSF and propose a novel model termed PETformer.

- We introduce the PET architecture, enabling direct modeling of relationships between historical and future sequences in LTSF tasks through attention mechanisms, resulting in en-

---

[1] To the best of our knowledge, we are the first to introduce the Feature Head technique in LTSF tasks.

hanced predictive performance. The design of the token-wise predictor in PET architecture significantly reduces the model's parameter count.

- Extensive experiments demonstrate that PETformer achieves state-of-the-art performance on eight public datasets for LTSF, surpassing all currently available models.

## 2 RELATED WORK

**Long-term time series forecasting**   In the domain of LTSF, traditional time series prediction methods, such as ARIMA (Box & Pierce, 1970), VAR (Kilian & Lütkepohl, 2017), DeepAR (Salinas et al., 2020), ConvLSTM (SHI et al., 2015), LSTnet (Lai et al., 2018), TCN (Bai et al., 2018; Sen et al., 2019), have shown limitations when dealing with extended look-back and forecasting horizons. The Transformer model, harnessing the self-attention mechanism, inherently possesses the capability to model long-term dependencies (Vaswani et al., 2017). Consequently, a considerable body of work has been dedicated to adapting the Transformer architecture for LTSF tasks, including LogTrans (Li et al., 2019), Informer (Zhou et al., 2021), Pyraformer (Liu et al., 2021a), Autoformer (Wu et al., 2021), and Fedformer (Zhou et al., 2022).

However, the applicability of the Transformer to LTSF tasks has been questioned by Dlinear (Zeng et al., 2023), which introduced a channel-independent strategy that significantly improved the performance of simple models in LTSF. The emergence of channel-independent strategies has also stimulated the development of numerous channel-independent-based methods (Das et al., 2023; Wang et al., 2023b; Han et al., 2023). PatchTST (Nie et al., 2023) has also adopted this strategy and introduced patch techniques from computer vision (Dosovitskiy et al., 2020; Liu et al., 2021b) to effectively enhance the performance of Transformer in LTSF, thereby providing strong evidence that the Transformer still excels in LTSF. In this paper, we are dedicated to further exploring the performance of the Transformer in the context of LTSF.

**Placeholder-enhanced techniques**   The Placeholder technique shares similarities with the currently popular Masking technique in the unsupervised pretraining domain. BERT initially introduced the Masking technique in natural language processing tasks, employing bidirectional Transformer to predict randomly masked text segments (Devlin et al., 2018). Subsequently, this technique has witnessed significant development in the computer vision domain (He et al., 2022; Bao et al., 2022; Xie et al., 2022). In the time series domain, attempts have also been made to leverage masking for unsupervised pretraining learning (Zerveas et al., 2021; Dong et al., 2023). It is worth noting that these techniques primarily focus on unsupervised pretraining for representation learning. In contrast to unsupervised learning approaches where known sequences are marked at random positions, this paper masks unknown future sequences for prediction, transforming random masking into semantically explicit placeholders.

## 3 MODEL ARCHITECTURE

The main architecture of PETformer, as depicted in Figure 2, takes input $X = \left\{ x_t^1, x_t^2, \cdots, x_t^d \right\}_{t=1}^l$, where $l$ signifies the length of the historical look-back window. Initially, it separates $X$ into $d$ independent channel sequences, resulting in $X' = \left\{ X^1, X^2, \cdots, X^d \right\}$. Subsequently, the independent sequences $X^i$ are partitioned into patches of length $w$ and transformed into input embeddings, generating $n$ tokens. These tokens are then input into the placeholder-enhanced Transformer encoder for feature extraction. This process yields $m$ tokens for each independent sequence, amounting to a total of $d \times m$ tokens after feature extraction from all $d$ channels. The $d \times m$ tokens are then transposed to obtain $m \times d$ tokens, which undergo the inter-channel interaction module for inter-channel feature extraction. Finally, the tokens are fed into the token-wise predictor, which employs parameter-sharing linear layer for individual patch output prediction. After concatenation, the final output $Y = \left\{ y_t^1, y_t^2, \cdots, y_t^d \right\}_{t=1}^h$ is obtained, where $h$ refers to the length of the forecast horizon. We will now provide detailed explanations of each module below.

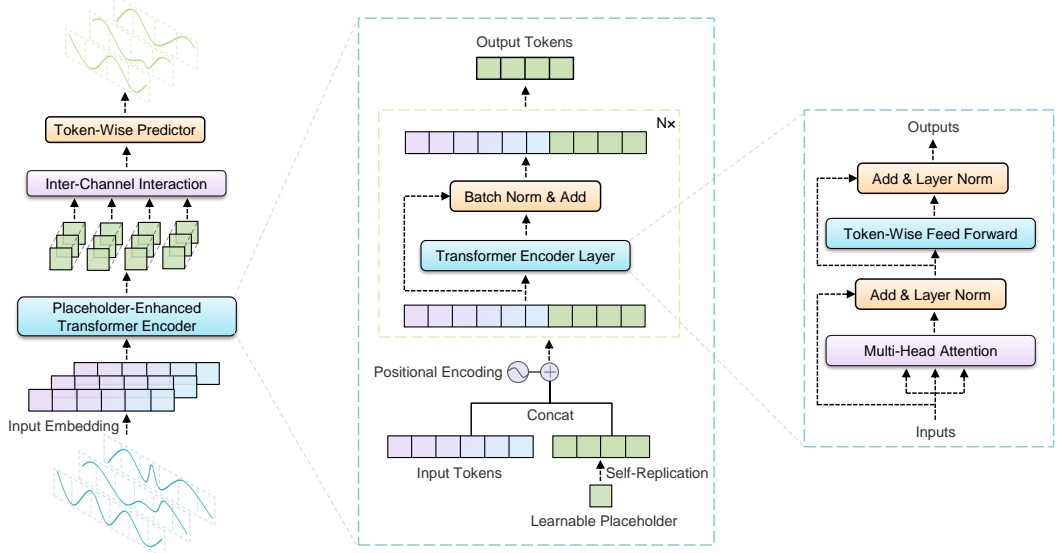

Figure 2: The PETformer model architecture.

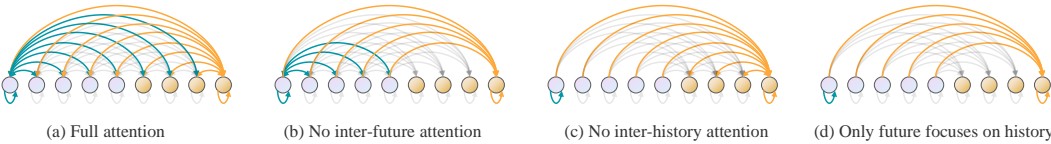

| (a) Full attention | (b) No inter-future attention | (c) No inter-history attention | (d) Only future focuses on history |

Figure 3: Different attention modes between history and future.

## 3.1 SUB-SEQUENCE DIVISION AND TOKENIZATION

For the given historical data, each independent channel sequence $X^i \in \mathbb{R}^l$ is divided into $n$ sub-sequences $X_{patched}^i \in \mathbb{R}^{n \times w}$, where $n$ is determined by the sub-sequence window length $w$ and the stride length $s$, i.e., $n = \lfloor \frac{l-w}{s} + 1 \rfloor$. Subsequently, each sub-sequence is mapped to a $d_{model}$ dimensional vector through a parameter-sharing linear layer, resulting in $X_{token}^i \in \mathbb{R}^{n \times d_{model}}$.

For the future data to be predicted, the same tokenization strategy is employed. A learnable placeholder $p \in \mathbb{R}^{d_{model}}$ is initially initialized, representing a future sub-sequence of length $w$. Subsequently, for a prediction horizon of $h$ time steps, $m$ placeholders are needed to represent the future, yielding $Y_{token}^i \in \mathbb{R}^{m \times d_{model}}$. In this case, $m = \lfloor \frac{h-w}{s} + 1 \rfloor$.

## 3.2 PLACEHOLDER-ENHANCED TRANSFORMER ENCODER

The placeholder-enhanced Transformer encoder takes an independent channel token sequence as input. Initially, a learnable placeholder is replicated $m$ times and concatenated with the $n$ tokens of input to form a sequence of $n+m$ tokens, which is then augmented with learnable position encoding. In this sequence, the first $n$ tokens encompass historical data, while the last $m$ tokens signify the future information to be learned, thereby facilitating direct interaction between historical and future information. The encoder contains $N$ feature extraction blocks, wherein the token sequence in each block undergoes intra-channel token feature interaction through a single Transformer encoder layer, succeeded by batch normalization (Ioffe & Szegedy, 2015) and residual connection (He et al., 2016). The Transformer encoder layer here is consistent with the vanilla Transformer (Vaswani et al., 2017). Finally, the encoder outputs the last $m$ tokens, which contain learned future prediction information.

Although the placeholders provide future-aware prior knowledge, they are not real future data, and whether more interaction should be done between historical data and placeholders is a question worth exploring. To this end, we explores four different attention modes (see Figure 3):

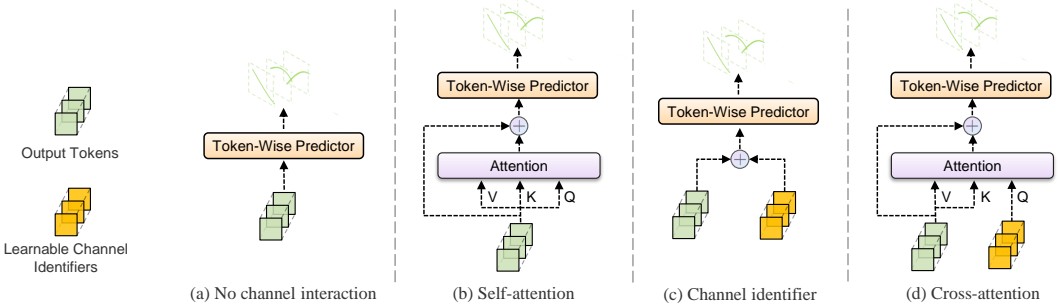

Figure 4: Different inter-channel interactions.

- **Full attention (FA):** This constitutes the standard operation, wherein bidirectional attention is conducted both within and between historical data and placeholders.

- **No inter-future attention (NIFA):** Historical data interact with each other and provide information to the placeholders. However, the placeholders do not interact with each other, nor do they provide information to the historical data..

- **No inter-history attention (NIHA):** Conversely, historical data can be kept unchanged, allowing only placeholders to interact with each other and continuously learn future information from the existing historical data.

- **Only future focuses on history (OFFH):** This configuration combines both NIFA and NIHA modes, where each placeholder independently focuses on historical information.

## 3.3 INTER-CHANNEL INTERACTION

Although Dlinear (Zeng et al., 2023) and PatchTST (Nie et al., 2023) have demonstrated that the simple channel separation strategy is sufficient to achieve high-level performance, considering the inter-channel dependencies might prove advantageous in predicting future information for certain scenarios. Therefore, we have incorporated an optional inter-channel feature extraction module into PETformer. Specifically, this work explores several methods for extracting inter-channel features (see Figure 4):

- **No Channel Interaction (NCI):** This strategy entails channel independence without additional inter-channel interactions, capturing potential dependencies between channels solely through the shared parameters within the Transformer encoder.

- **Self-Attention (SA):** Building upon the NCI approach, SA utilizes self-attention mechanisms to extract inter-channel features.

- **Channel Identifier (CI):** Extending from NCI, the CI strategy leverages channel identifiers to distinguish different channels more effectively. This enables the model to discern distinct channel patterns. Channel identifier technology has recently emerged as a substitute for complex graph neural networks in spatial modeling (Shao et al., 2022).

- **Cross-Attention (CA):** CA represents a fusion of the SA and CI strategies. It uses channel identifiers as queries, replacing the original self-attention queries, thereby modeling inter-channel dependencies through cross-attention mechanisms.

## 3.4 INSTANCE NORMALIZATION AND LOSS FUCTION

**Instance normalization**   Besides the global preprocessing normalization of data, we also employ RevIN (Kim et al., 2021; Ulyanov et al., 2016) to tackle the distribution shift issue in time-series data between training and test sets. RevIN normalizes each individual sample prior to inputting it into the model and denormalizes it after obtaining the output from the model.

**Loss function**   We utilize Smooth L1 loss (Girshick, 2015), which amalgamates the advantages of both L1 loss and L2 loss. This combined approach offers a more stable gradient, mitigating the

likelihood of gradient explosion or decay during the training phase. In this work, the loss function is defined as follows:

$$\mathcal{L}(Y, \hat{Y}) = \frac{1}{d \cdot h} \sum_{t=1}^{h} \sum_{i=1}^{d} \rho_{\text{smooth}}(y_t^i - \hat{y}_t^i),$$

where

$$\rho_{\text{smooth}}(x) = \begin{cases} 0.5x^2 & \text{for } |x| < 1 \\ |x| - 0.5 & \text{otherwise.} \end{cases}$$

## 4 EXPERIMENTS

In this section, we present the main results of PETformer on multivariate and univariate time series forecasting tasks, as well as ablation experiments and model analyses. More details and other supplementary experiments are available in Appendix A.

Table 1: Summary of Dataset Characteristics

| Datasets | ETTh1 | ETTh2 | ETTm1 | ETTm2 | Electricity | ILI | Traffic | Weather |
|---|---|---|---|---|---|---|---|---|
| Dimension | 7 | 7 | 7 | 7 | 321 | 7 | 862 | 21 |
| Frequency | 1 hour | 1 hour | 15 mins | 15 mins | 1 hour | 7 days | 1 hour | 10 mins |
| Length | 17,420 | 17,420 | 69,680 | 69,680 | 26,304 | 966 | 52,696 | 52,696 |

**Dataset** We conducted extensive experiments on eight widely used public datasets, including ETTs (ETTh1, ETTh2, ETTm1, ETTm2) (Zhou et al., 2021), Electricity, ILI, Weather, and Traffic (Wu et al., 2021), covering energy, transportation, medical, and weather domains. Table 1 presents key characteristics of the eight datasets.

**Baselines and metrics** We choose state-of-the-art and representative LTSF models as our baselines, comprising Transformer-based models like PatchTST (Nie et al., 2023), Crossformer (Zhang & Yan, 2023), FEDformer (Zhou et al., 2022), Autoformer (Wu et al., 2021), and Informer (Zhou et al., 2021), in addition to non-Transformer-based models such as Dlinear (Zeng et al., 2023) and Micn (Wang et al., 2023a). To assess the performance of these models, we employ widely used evaluation metrics: Mean Squared Error (MSE) and Mean Absolute Error (MAE). In every table presented in this paper, the best results are emphasized in **bold**, whereas the second are underlined.

### 4.1 MAIN RESULTS

Table 2 presents the outcomes of multivariate long-term forecasting. Overall, PETformer achieves state-of-the-art performance on all prediction step settings for the eight datasets, outperforming all baseline methods. On average, compared to the most recent advanced model PatchTST, PETformer achieves a 4.7% MSE improvement and a 3.7% MAE improvement. Notably, for Dlinear, which has questioned the efficacy of Transformer in LTSF, PETformer achieves a 24.3% MSE improvement and a 14.9% MAE improvement.

Furthermore, PETformer still achieves the best performance on the full ETT datasets in univariate prediction scenarios, outperforming all baselines (Table 3). On average, compared to PatchTST, PETformer achieves a 4.9% MSE improvement and a 2.3% MAE improvement. This demonstrates that the PETformer's designs indeed bring more useful prior knowledge to time series prediction tasks.

### 4.2 ABLATION STUDIES AND ANALYSES

We conducted ablation experiments on three datasets of varying scales and numbers of variables (ETTh1, Weather, and Traffic) to thoroughly assess the effectiveness of the proposed methods.

#### 4.2.1 PLACEHOLDER-ENHANCED TECHNIQUE

Table 4 presents the outcomes of the ablation study for PET. The native Encoder-Decoder architecture in the Transformer yields the poorest performance, corroborating that cross self-attention

Table 2: Multivariate long-term series forecasting results. PETformer employs a look-back window length of $l = 72$ for the ILI dataset and $l = 720$ for the remaining datasets. The forecast horizon $h \in \{24, 36, 48, 60\}$ is set for the ILI datasets and $h \in \{96, 192, 336, 720\}$ is set for the others.

| Models | | PETformer (ours) | | PatchTST (2023) | | Dlinear (2023) | | MICN (2023) | | Crossformer (2023) | | FEDformer* (2022) | | Autoformer* (2021) | | Informer* (2021) | |
|---|---|---|---|---|---|---|---|---|---|---|---|---|---|---|---|---|---|
| Metric | | MSE | MAE | MSE | MAE | MSE | MAE | MSE | MAE | MSE | MAE | MSE | MAE | MSE | MAE | MSE | MAE |
| ETTh1 | 96 | **0.347** | **0.377** | 0.376 | 0.408 | 0.378 | 0.402 | 0.404 | 0.429 | 0.380 | 0.419 | 0.376 | 0.415 | 0.435 | 0.446 | 0.941 | 0.769 |
| | 192 | **0.390** | **0.404** | 0.416 | 0.423 | 0.415 | 0.425 | 0.475 | 0.484 | 0.419 | 0.445 | 0.423 | 0.446 | 0.456 | 0.457 | 1.007 | 0.786 |
| | 336 | **0.419** | **0.418** | 0.425 | 0.440 | 0.447 | 0.448 | 0.482 | 0.489 | 0.438 | 0.451 | 0.444 | 0.462 | 0.486 | 0.487 | 1.038 | 0.784 |
| | 720 | **0.437** | **0.449** | 0.448 | 0.470 | 0.480 | 0.489 | 0.599 | 0.576 | 0.508 | 0.514 | 0.469 | 0.492 | 0.515 | 0.517 | 1.144 | 0.857 |
| ETTh2 | 96 | **0.272** | **0.329** | 0.275 | 0.338 | 0.282 | 0.346 | 0.289 | 0.354 | 0.383 | 0.420 | 0.332 | 0.374 | 0.332 | 0.368 | 1.549 | 0.952 |
| | 192 | **0.338** | **0.374** | 0.338 | 0.378 | 0.350 | 0.396 | 0.408 | 0.444 | 0.421 | 0.450 | 0.407 | 0.446 | 0.426 | 0.434 | 3.792 | 1.542 |
| | 336 | **0.328** | **0.380** | 0.329 | 0.380 | 0.410 | 0.437 | 0.547 | 0.516 | 0.449 | 0.459 | 0.4 | 0.447 | 0.477 | 0.479 | 4.215 | 1.642 |
| | 720 | 0.401 | 0.439 | 0.379 | 0.422 | 0.587 | 0.544 | 0.834 | 0.688 | 0.472 | 0.497 | 0.412 | 0.469 | 0.453 | 0.490 | 3.656 | 1.619 |
| ETTm1 | 96 | **0.282** | **0.325** | 0.293 | 0.342 | 0.306 | 0.345 | 0.301 | 0.352 | 0.295 | 0.350 | 0.326 | 0.390 | 0.51 | 0.492 | 0.626 | 0.560 |
| | 192 | **0.318** | **0.349** | 0.328 | 0.365 | 0.335 | 0.365 | 0.344 | 0.380 | 0.339 | 0.381 | 0.365 | 0.415 | 0.514 | 0.495 | 0.725 | 0.619 |
| | 336 | **0.348** | **0.372** | 0.362 | 0.394 | 0.373 | 0.391 | 0.379 | 0.401 | 0.419 | 0.432 | 0.392 | 0.425 | 0.51 | 0.492 | 1.005 | 0.741 |
| | 720 | **0.404** | **0.403** | 0.414 | 0.420 | 0.422 | 0.422 | 0.429 | 0.429 | 0.579 | 0.551 | 0.446 | 0.458 | 0.527 | 0.493 | 1.133 | 0.845 |
| ETTm2 | 96 | **0.160** | **0.248** | 0.163 | 0.255 | 0.164 | 0.259 | 0.177 | 0.274 | 0.296 | 0.352 | 0.18 | 0.271 | 0.205 | 0.293 | 0.355 | 0.462 |
| | 192 | **0.217** | **0.288** | 0.221 | 0.292 | 0.233 | 0.314 | 0.236 | 0.310 | 0.342 | 0.385 | 0.252 | 0.318 | 0.278 | 0.336 | 0.595 | 0.586 |
| | 336 | 0.274 | **0.326** | 0.270 | 0.329 | 0.291 | 0.355 | 0.299 | 0.350 | 0.410 | 0.425 | 0.324 | 0.364 | 0.343 | 0.379 | 1.27 | 0.871 |
| | 720 | **0.345** | **0.376** | 0.347 | 0.378 | 0.407 | 0.433 | 0.421 | 0.434 | 0.563 | 0.538 | 0.41 | 0.420 | 0.414 | 0.419 | 3.001 | 1.267 |
| Electricity | 96 | **0.128** | **0.220** | 0.130 | 0.223 | 0.133 | 0.230 | 0.151 | 0.260 | 0.198 | 0.292 | 0.186 | 0.302 | 0.196 | 0.313 | 0.304 | 0.393 |
| | 192 | **0.144** | **0.236** | 0.147 | 0.240 | 0.147 | 0.244 | 0.165 | 0.276 | 0.266 | 0.330 | 0.197 | 0.311 | 0.211 | 0.324 | 0.327 | 0.417 |
| | 336 | **0.159** | **0.252** | 0.164 | 0.257 | 0.162 | 0.261 | 0.183 | 0.291 | 0.343 | 0.377 | 0.213 | 0.328 | 0.214 | 0.327 | 0.333 | 0.422 |
| | 720 | **0.195** | **0.286** | 0.203 | 0.292 | 0.196 | 0.294 | 0.201 | 0.312 | 0.398 | 0.422 | 0.233 | 0.344 | 0.236 | 0.342 | 0.351 | 0.427 |
| ILI | 24 | **1.204** | **0.687** | 1.356 | 0.732 | 2.000 | 0.987 | 2.483 | 1.058 | 3.217 | 1.198 | 2.624 | 1.095 | 2.906 | 1.182 | 4.657 | 1.449 |
| | 36 | 1.246 | 0.709 | 1.244 | 0.705 | 2.202 | 1.026 | 2.370 | 0.987 | 3.136 | 1.199 | 2.516 | 1.021 | 2.585 | 1.038 | 4.65 | 1.463 |
| | 48 | **1.446** | **0.760** | 1.604 | 0.791 | 2.278 | 1.059 | 2.371 | 1.007 | 3.331 | 1.236 | 2.505 | 1.041 | 3.024 | 1.145 | 5.004 | 1.542 |
| | 60 | **1.430** | **0.774** | 1.648 | 0.860 | 2.478 | 1.111 | 2.513 | 1.055 | 3.609 | 1.265 | 2.742 | 1.122 | 2.761 | 1.114 | 5.071 | 1.543 |
| Traffic | 96 | **0.357** | **0.240** | 0.367 | 0.253 | 0.385 | 0.269 | 0.445 | 0.295 | 0.487 | 0.274 | 0.576 | 0.359 | 0.597 | 0.371 | 0.733 | 0.410 |
| | 192 | **0.376** | **0.248** | 0.382 | 0.259 | 0.395 | 0.273 | 0.461 | 0.302 | 0.497 | 0.279 | 0.61 | 0.380 | 0.607 | 0.382 | 0.777 | 0.435 |
| | 336 | **0.392** | **0.255** | 0.396 | 0.267 | 0.409 | 0.281 | 0.483 | 0.307 | 0.517 | 0.285 | 0.608 | 0.375 | 0.623 | 0.387 | 0.776 | 0.434 |
| | 720 | **0.430** | **0.276** | 0.433 | 0.287 | 0.449 | 0.305 | 0.527 | 0.310 | 0.584 | 0.323 | 0.621 | 0.375 | 0.639 | 0.395 | 0.827 | 0.466 |
| Weather | 96 | 0.146 | **0.186** | 0.147 | 0.198 | 0.169 | 0.231 | 0.167 | 0.231 | **0.144** | 0.208 | 0.238 | 0.314 | 0.249 | 0.329 | 0.354 | 0.405 |
| | 192 | **0.190** | **0.229** | 0.190 | 0.241 | 0.213 | 0.273 | 0.212 | 0.271 | 0.192 | 0.263 | 0.275 | 0.329 | 0.325 | 0.370 | 0.419 | 0.434 |
| | 336 | **0.241** | **0.271** | 0.243 | 0.284 | 0.260 | 0.314 | 0.275 | 0.337 | 0.246 | 0.306 | 0.339 | 0.377 | 0.351 | 0.391 | 0.583 | 0.543 |
| | 720 | 0.314 | **0.323** | 0.305 | 0.328 | 0.315 | 0.353 | 0.312 | 0.349 | 0.318 | 0.361 | 0.389 | 0.409 | 0.415 | 0.426 | 0.916 | 0.705 |
| **Avg.** | | **0.427** | **0.369** | 0.448 | 0.383 | 0.565 | 0.434 | 0.623 | 0.455 | 0.756 | 0.490 | 0.651 | 0.472 | 0.713 | 0.497 | 1.629 | 0.825 |

∗ denotes that the data originates from PatchTST (Nie et al., 2023).

Table 3: Univariate long-term series forecasting results.

| Models | | PETformer (ours) | | PatchTST (2023) | | Dlinear (2023) | | MICN (2023) | | FEDformer* (2022) | | Autoformer* (2021) | | Informer* (2021) | |
|---|---|---|---|---|---|---|---|---|---|---|---|---|---|---|---|
| Metric | | MSE | MAE | MSE | MAE | MSE | MAE | MSE | MAE | MSE | MAE | MSE | MAE | MSE | MAE |
| ETTh1 | 96 | **0.052** | **0.174** | 0.055 | 0.179 | 0.056 | 0.185 | 0.056 | 0.179 | 0.079 | 0.215 | 0.071 | 0.206 | 0.193 | 0.377 |
| | 192 | **0.066** | **0.201** | 0.070 | 0.202 | 0.077 | 0.218 | 0.071 | 0.203 | 0.104 | 0.245 | 0.114 | 0.262 | 0.217 | 0.395 |
| | 336 | **0.075** | **0.217** | 0.077 | 0.225 | 0.085 | 0.228 | 0.086 | 0.229 | 0.119 | 0.270 | 0.107 | 0.258 | 0.202 | 0.381 |
| | 720 | **0.079** | **0.225** | 0.087 | 0.232 | 0.159 | 0.322 | 0.150 | 0.316 | 0.142 | 0.299 | 0.126 | 0.283 | 0.183 | 0.355 |
| ETTh2 | 96 | **0.120** | 0.272 | 0.127 | 0.273 | 0.122 | **0.269** | 0.128 | 0.271 | 0.128 | 0.271 | 0.153 | 0.306 | 0.213 | 0.373 |
| | 192 | **0.156** | **0.316** | 0.168 | 0.328 | 0.166 | 0.320 | 0.175 | 0.328 | 0.185 | 0.330 | 0.204 | 0.351 | 0.227 | 0.387 |
| | 336 | **0.164** | **0.328** | 0.172 | 0.338 | 0.188 | 0.349 | 0.192 | 0.354 | 0.231 | 0.378 | 0.246 | 0.389 | 0.242 | 0.401 |
| | 720 | **0.208** | **0.367** | 0.223 | 0.382 | 0.311 | 0.454 | 0.268 | 0.418 | 0.278 | 0.420 | 0.268 | 0.409 | 0.291 | 0.439 |
| ETTm1 | 96 | **0.026** | **0.120** | 0.026 | 0.121 | 0.028 | 0.125 | 0.027 | 0.123 | 0.033 | 0.140 | 0.056 | 0.183 | 0.109 | 0.277 |
| | 192 | **0.039** | **0.148** | 0.039 | 0.150 | 0.045 | 0.156 | 0.043 | 0.154 | 0.058 | 0.186 | 0.081 | 0.216 | 0.151 | 0.310 |
| | 336 | **0.052** | **0.171** | 0.053 | 0.173 | 0.055 | 0.175 | 0.052 | 0.173 | 0.084 | 0.231 | 0.076 | 0.218 | 0.427 | 0.591 |
| | 720 | **0.070** | **0.201** | 0.072 | 0.206 | 0.076 | 0.209 | 0.075 | 0.206 | 0.102 | 0.250 | 0.110 | 0.267 | 0.438 | 0.586 |
| ETTm2 | 96 | 0.062 | 0.181 | 0.063 | 0.186 | **0.061** | **0.179** | 0.063 | 0.183 | 0.067 | 0.198 | 0.065 | 0.189 | 0.088 | 0.225 |
| | 192 | **0.087** | **0.222** | 0.094 | 0.230 | 0.095 | 0.234 | 0.091 | 0.225 | 0.102 | 0.245 | 0.118 | 0.256 | 0.132 | 0.283 |
| | 336 | **0.118** | **0.264** | 0.120 | 0.265 | 0.122 | 0.265 | 0.121 | 0.265 | 0.130 | 0.279 | 0.154 | 0.305 | 0.180 | 0.336 |
| | 720 | **0.163** | **0.317** | 0.171 | 0.321 | 0.173 | 0.324 | 0.172 | 0.317 | 0.178 | 0.325 | 0.182 | 0.335 | 0.300 | 0.435 |
| **Avg.** | | **0.096** | **0.233** | 0.101 | 0.238 | 0.114 | 0.251 | 0.111 | 0.247 | 0.126 | 0.268 | 0.133 | 0.277 | 0.225 | 0.384 |

∗ denotes that the data originates from PatchTST (Nie et al., 2023).

may lead to the loss of temporal dependency transmission in LTSF (Li et al., 2023). The Flattening technique substantially enhances Transformer performance, aligning with observations in PatchTST (Nie et al., 2023). Although, to the best of our knowledge, the Feature Head technique has not been applied in the LTSF domain, our research indicates that its performance surpasses the Flattening technique. As anticipated, our PET approach achieves the best results, regardless of the attention mode in PET, demonstrating the superiority of the PET method. Furthermore, the Full At-

tention (FA) mode outperforms the other three modes, indirectly substantiating that allowing more historical and future data to interact directly at the same level of attention is advantageous.

Table 4: Ablation study of the PET. Various application techniques for Transformer, as depicted in Figure 1, are incorporated. In the case of PET, we evaluated different attention modes between historical and future data, as illustrated in Figure 3.

| Models | | Enc-dec | | Flattening | | Feature Head | | PET/FA† | | PET/NIFA | | PET/NIHA | | PET/OFFH | |
|---|---|---|---|---|---|---|---|---|---|---|---|---|---|---|---|
| Metric | | MSE | MAE | MSE | MAE | MSE | MAE | MSE | MAE | MSE | MAE | MSE | MAE | MSE | MAE |
| ETTh1 | 96 | 0.399 | 0.409 | 0.377 | 0.402 | 0.350 | 0.382 | 0.348 | 0.379 | **0.347** | **0.378** | 0.352 | 0.381 | 0.349 | 0.379 |
| | 192 | 0.435 | 0.439 | 0.416 | 0.427 | 0.391 | 0.408 | 0.389 | **0.403** | 0.390 | 0.403 | **0.388** | 0.403 | 0.390 | 0.403 |
| | 336 | 0.472 | 0.456 | 0.449 | 0.447 | 0.425 | 0.434 | **0.419** | 0.420 | 0.422 | **0.419** | 0.425 | 0.422 | 0.425 | 0.421 |
| | 720 | 0.513 | 0.496 | 0.469 | 0.476 | 0.445 | 0.462 | **0.435** | **0.447** | 0.450 | 0.456 | 0.443 | 0.453 | 0.453 | 0.457 |
| Weather | 96 | 0.196 | 0.257 | **0.145** | 0.189 | 0.145 | 0.186 | 0.145 | **0.184** | 0.145 | 0.185 | 0.147 | 0.186 | 0.147 | 0.187 |
| | 192 | 0.239 | 0.289 | 0.193 | 0.235 | 0.191 | 0.231 | **0.190** | **0.228** | 0.190 | 0.229 | 0.191 | 0.230 | 0.191 | 0.230 |
| | 336 | 0.299 | 0.331 | 0.245 | 0.278 | 0.242 | 0.273 | **0.241** | **0.270** | 0.241 | 0.270 | 0.242 | 0.271 | 0.241 | 0.270 |
| | 720 | 0.371 | 0.388 | 0.323 | 0.334 | 0.318 | 0.328 | 0.312 | 0.322 | 0.314 | 0.323 | 0.311 | 0.322 | **0.311** | **0.322** |
| Traffic | 96 | 0.400 | 0.305 | 0.368 | 0.252 | 0.362 | 0.252 | 0.361 | 0.247 | **0.361** | **0.246** | 0.375 | 0.259 | 0.371 | 0.256 |
| | 192 | 0.419 | 0.313 | 0.387 | 0.261 | 0.389 | 0.268 | **0.377** | **0.253** | 0.380 | 0.254 | 0.384 | 0.259 | 0.385 | 0.259 |
| | 336 | 0.422 | 0.316 | 0.402 | 0.269 | 0.407 | 0.278 | **0.394** | **0.263** | 0.395 | 0.263 | 0.398 | 0.266 | 0.399 | 0.268 |
| | 720 | 0.466 | 0.336 | 0.443 | 0.290 | 0.451 | 0.303 | **0.430** | **0.282** | 0.433 | 0.283 | 0.434 | 0.287 | 0.438 | 0.289 |
| **Avg.** | | 0.386 | 0.361 | 0.351 | 0.322 | 0.343 | 0.317 | **0.337** | **0.308** | 0.339 | 0.309 | 0.341 | 0.312 | 0.342 | 0.312 |

† denotes the default mode in this work.

Additionally, the advantages of the PET architecture extend beyond prediction accuracy and also manifest in parameter efficiency and computational performance. PatchTST utilizes the Flattening technique, wherein $n$ tokens of dimension $d$ output from the Transformer encoder would be flatten into a single token, followed by an linear layer ($n \times d$, $h$) for final prediction. The prediction phase necessitates $n \times d \times h$ learnable parameters, which is quite substantial. In contrast, the PET architecture's token-wise prediction head requires significantly fewer parameters, namely $d \times w$.

Table 5 presents runtime measurements with different models. Compared to PatchTST, which requires a hefty 8.3 million parameters for its heavy flattening-prediction head, PETformer's lightweight token-wise prediction head demands only 6.19 thousand parameters, representing a reduction of over 99%. In terms of the proportion of prediction head parameters relative to the total model parameters, PETformer accounts for just 1.3%, as opposed to PatchTST's

Table 5: Comparison of runtime measurements with different models. These results are based on the in-720-out-720 task of the ETTm1 dataset, using a Transformer with a hidden dimension of 128 and a layer count of 3.

| Models | PETformer | PatchTST | Autoformer | Informer |
|---|---|---|---|---|
| Train Time (s/epoch) | 22.2 | 35.1 | 55.4 | 43.1 |
| MACs (MMac) | 96.7 | 308 | 441 | 358 |
| Pred. Layer Parameters | 6.19k | 8.30M | 903 | 903 |
| Total Parameters | 480k | 8.71M | 602k | 703k |
| Max Memory (MB) | 475 | 2,231 | 3,324 | 1,352 |

nearly 95%. Allowing more learnable parameters to influence the feature extraction module of the Transformer, rather than the linear prediction head, may be a key reason for PETformer outperforming PatchTST. Moreover, in metrics such as training time and maximum memory usage, PETformer outperforms other models, highlighting the high computational efficiency of the PET architecture.

### 4.2.2 LONG SUB-SEQUENCE DIVISION AND MULTI-CHANNEL SEPARATION AND INTERACTION

Table 6 presents the outcomes of the ablation study for LSD and MSI. The point-wise attention approach yielded the poorest performance. In contrast, the LSD approach demonstrated significantly better results. Notably, as the window size of the sub-sequence incrementally increased, the LTSF performance continued to improve. Although the performance improvement due to the growth of $w$ has not yet saturated, larger values of $w$ cannot be explored in this work since $w = 48$ already represents the greatest common divisor of $h \in \{96, 192, 336, 720\}$. While the concept of LSD is not novel (i.e., it inherits the patch strategy from PatchTST), we have substantiated that larger window sub-sequence divisions offer richer semantic information within the Transformer architecture, thereby directly influencing the performance of LTSF.

Table 6: Ablation study of the LSD and MSI. For the LSD, a comparison of performance between using single points and sub-sequences (various window lengths $w \in \{6, 12, 24, 48\}$) as input granularity is presented. For the MSI, The direct channel mixing (DCM) approach is compared with the MSI approach (various strategies illustrated in Figure 4). The '-' symbol denotes out-of-memory issues encountered in our experimental environment.

| Models | | Ponit | | LSD/w=6 | | LSD/w=12 | | LSD/w=24 | | LSD/w=48 | | DCM | | MSI/NCI | | MSI/SA | | MSI/CI | | MSI/CA | |
|---|---|---|---|---|---|---|---|---|---|---|---|---|---|---|---|---|---|---|---|---|---|
| Metric | | MSE | MAE | MSE | MAE | MSE | MAE | MSE | MAE | MSE | MAE | MSE | MAE | MSE | MAE | MSE | MAE | MSE | MAE | MSE | MAE |
| ETTh1 | 96 | 0.910 | 0.424 | 0.435 | 0.424 | 0.361 | 0.390 | 0.349 | 0.380 | **0.347** | **0.377** | 0.409 | 0.435 | 0.348 | 0.379 | **0.348** | 0.379 | 0.349 | 0.378 | 0.349 | **0.378** |
| | 192 | 0.920 | 0.472 | 0.517 | 0.472 | 0.406 | 0.416 | 0.392 | 0.405 | **0.390** | **0.404** | 0.470 | 0.476 | 0.389 | 0.403 | **0.389** | **0.403** | 0.394 | 0.407 | 0.393 | 0.407 |
| | 336 | 0.815 | 0.539 | 0.636 | 0.539 | 0.435 | 0.430 | 0.424 | 0.425 | **0.419** | **0.418** | 0.512 | 0.503 | 0.419 | 0.419 | 0.419 | 0.419 | **0.415** | **0.417** | 0.426 | 0.426 |
| | 720 | 0.832 | 0.604 | 0.719 | 0.604 | 0.461 | 0.468 | 0.456 | 0.458 | **0.437** | **0.449** | 0.626 | 0.574 | 0.444 | 0.452 | **0.435** | **0.447** | 0.454 | 0.458 | 0.453 | 0.461 |
| Weather | 96 | 0.159 | 0.202 | 0.159 | 0.200 | 0.149 | 0.189 | 0.146 | 0.186 | **0.146** | **0.186** | 0.147 | 0.195 | 0.145 | 0.185 | 0.145 | **0.184** | 0.145 | 0.185 | **0.144** | **0.184** |
| | 192 | 0.206 | 0.243 | 0.206 | 0.244 | 0.193 | 0.233 | **0.189** | 0.229 | 0.190 | **0.229** | 0.198 | 0.245 | 0.189 | **0.228** | 0.189 | **0.228** | **0.188** | 0.229 | 0.189 | 0.228 |
| | 336 | 0.255 | 0.289 | 0.256 | 0.284 | 0.244 | 0.274 | **0.241** | **0.271** | **0.241** | **0.271** | 0.255 | 0.291 | 0.241 | **0.270** | 0.241 | **0.270** | 0.242 | 0.272 | **0.241** | 0.271 |
| | 720 | 0.323 | 0.333 | 0.326 | 0.340 | 0.315 | 0.327 | **0.312** | **0.324** | 0.314 | **0.323** | 0.321 | 0.340 | 0.315 | 0.323 | 0.313 | **0.322** | **0.311** | **0.322** | 0.314 | 0.324 |
| Traffic | 96 | - | - | 0.437 | 0.316 | 0.378 | 0.251 | 0.365 | 0.243 | **0.357** | **0.240** | 1.530 | 0.621 | **0.357** | **0.240** | 0.363 | 0.249 | 0.359 | 0.243 | 0.365 | 0.248 |
| | 192 | - | - | 0.485 | 0.356 | 0.392 | 0.257 | 0.382 | 0.250 | **0.376** | **0.248** | 1.516 | 0.614 | **0.376** | **0.248** | 0.377 | 0.253 | **0.375** | 0.249 | 0.386 | 0.257 |
| | 336 | - | - | 0.492 | 0.345 | 0.413 | 0.270 | 0.397 | 0.258 | **0.392** | **0.255** | 1.546 | 0.631 | **0.392** | **0.255** | 0.393 | 0.262 | 0.393 | 0.257 | 0.400 | 0.265 |
| | 720 | - | - | 0.485 | 0.327 | 0.483 | 0.326 | 0.435 | 0.278 | **0.430** | **0.276** | 1.566 | 0.644 | **0.430** | **0.276** | 0.430 | 0.282 | 0.431 | 0.278 | 0.438 | 0.285 |
| Avg. | | - | - | 0.429 | 0.371 | 0.352 | 0.319 | 0.341 | 0.309 | **0.337** | **0.306** | 0.758 | 0.464 | 0.337 | **0.307** | **0.337** | 0.308 | 0.338 | 0.308 | 0.341 | 0.311 |

As for MSI, the DCM approach yielded the least favorable results in our experiments, suggesting that the DCM approach might severely disrupt intra-channel temporal dependencies within MTS. Conversely, the simple channel-independent strategy (i.e., NCI) significantly outperformed DCM, consistent with the observations of Dlinear and PatchTST. Although channel independence is potent, the absence of channel interaction in the context of multivariate time series forecasting appears to be a less desirable attribute. We further explored several channel interaction methods on top of the channel-independent strategy (i.e., first extracting temporal features through the channel-independent strategy and then extracting inter-variable features through channel interaction). Our observations include: (i) MSI results also substantially outperform DCM, demonstrating that extracting temporal features within channels before inter-variable feature extraction is a superior choice compared to directly extracting channel features. (ii) Several additional channel interaction methods did not yield significant advantages over channel independence, prompting further considerations as outlined below.

We hypothesize that this phenomenon arises due to MTS exhibiting varying numbers of channels, leading to intricate and challenging-to-estimate inter-channel relationships. For instance, in the ETTh1 dataset, which contains only 7 variables, inter-channel interaction can result in marginal performance improvements. Conversely, in the Traffic dataset with its 862 variables, the inter-channel relationships become too complex for the model to effectively handle. From this perspective, a channel-independent strategy may currently represent the optimal choice, aligning with the recent proliferation of channel-independent-based approaches. Nevertheless, this also underscores the ongoing challenges in devising technical solutions capable of accurately modeling such intricate relationships within multivariate time series. It points to potential avenues for future research aimed at effectively modeling the intricate interrelationships among multivariate channels.

## 5 CONCLUSION

In this paper, we have investigated the factors influencing the Transformer's performance in the LTSF domain, considering three potential perspectives: temporal continuity, information density, and multi-channel relationships. We introduced a comprehensive solution, PETformer, with its key design component, the Placeholder-enhanced Technique (PET), which not only enhances prediction accuracy but also improves computational efficiency. The experimental results conducted on eight public datasets have clearly demonstrated that PETformer outperforms existing models, achieving state-of-the-art performance. We have also conducted an extensive set of ablation experiments and a systematic analysis of the factors contributing to the success of PETformer. We posit that these insights can provide valuable guidance and inspiration for future research in the realm of time series tasks. In particular, the exploration of more effective modeling of relationships between multiple variables represents a promising and important direction for future research.

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

## A  APPENDIX

In this section, we present the experimental details of PETformer and provide additional supportive experiments to further demonstrate its effectiveness. The organization of this section is as follows:

- Appendix A.1 provides details on the datasets, baselines, configurations, and environments used in the experiments.
- Appendix A.2 presents more ablation experiments about RevIN and Smooth L1 loss.
- Appendix A.3 investigates the impact of the look-back window length on model performance.
- Appendix A.4 discusses the effect of channel separation on the model's ability to predict individual channels.
- Appendix A.5 presents the results of the robustness experiments conducted to assess the model's stability to random seed perturbations.
- Appendix A.6 showcases the results on the large datasets to demonstrate the model's predictive ability.

### A.1  EXPERIMENTAL DETAILS

#### A.1.1  DATASETS

We use the most popular multivariate datasets in LTSF, including ETT[2] (ETTh1, ETTh2, ETTm1, ETTm2), Electricity[3], ILI[4], Traffic[5], and Weather[6]. These datasets encompass various domains, such as energy, healthcare, transportation, and weather. The dimensions of each dataset range from 7 to 862, with frequencies ranging from 10 minutes to 7 days. The length of the datasets varies from

---

[2]https://github.com/zhouhaoyi/ETDataset

[3]https://archive.ics.uci.edu/ml/datasets/ElectricityLoadDiagrams20112014

[4]https://gis.cdc.gov/grasp/fluview/fluportaldashboard.html

[5]https://pems.dot.ca.gov/

[6]https://www.bgc-jena.mpg.de/wetter/

966 to 69,680 data points. We split all datasets into training, validation, and test sets in chronological order, using a ratio of 6:2:2 for the ETT dataset and 7:1:2 for the remaining datasets. For a more detailed discussion on the datasets, we recommend referring to the Autoformer paper (Wu et al., 2021).

### A.1.2 BASELINES

We choose state-of-the-art and the most representative LTSF models as our baselines, including both Transformer-based and non-Transformer-based models, as follows:

- PatchTST (Nie et al., 2023): the current state-of-the-art LTSF model as of July 2023. It utilizes channel-independent and patch techniques and achieves the highest performance by utilizing the native Transformer.
- Dlinear (Zeng et al., 2023): a highly insightful work that employs simple linear models and trend decomposition techniques, outperforming all Transformer-based models at the time. This work inspired us to reflect on the utility of Transformer in LTSF and indirectly led to the birth of PETformer in our study.
- Micn (Wang et al., 2023a): another non-Transformer model that enhances the performance of CNN models in LTSF through down-sampled convolution and isometric convolution, outperforming many Transformer-based models. This excellent work has been selected for oral presentation at ICLR 2023.
- Crossformer (Zhang & Yan, 2023): similar to PatchTST, it utilizes the patch technique commonly used in the computer vision domain. However, unlike PatchTST's independent channel design, it leverages cross-dimension dependency to enhance LTSF performance. This outstanding work has also been selected for oral presentation at ICLR 2023.
- FEDformer (Zhou et al., 2022): it employs trend decomposition and Fourier transformation techniques to improve the performance of Transformer-based models in LTSF. It was the best-performing Transformer-based model before Dlinear.
- Autoformer (Wu et al., 2021): it combines trend decomposition techniques with an auto-correlation mechanism, inspiring subsequent work such as FEDformer.
- Informer (Zhou et al., 2021): it proposes improvements to the Transformer model by utilizing a sparse self-attention mechanism and generative-style decoder, inspiring a series of subsequent Transformer-based LTSF models. This work was awarded Best Paper at AAAI 2021.

PETformer exhibited more robust performance with an ultra-long look-back window (i.e., $l = 720$), while the baseline models showed significant performance variations across different look-back window lengths. To fairly evaluate the performance of PETformer and baseline models, we conducted additional experiments with the other models using look-back window lengths of $l \in \{24, 48, 72, 144\}$ for the ILI dataset and $l \in \{96, 192, 336, 720\}$ for the other datasets, and selected the best result from these experiments as their final outcome in Tables 2 and 3.

We performed these experiments on their publicly available official repositories, prioritizing their default parameters and only modifying the look-back window length $l$ and the prediction horizon $h$. The official open-source codes for these baseline models are as follows:

- PatchTST: https://github.com/yuqinie98/patchtst
- Dlinear: https://github.com/cure-lab/LTSF-Linear
- Micn: https://github.com/wanghq21/MICN
- Crossformer: https://github.com/Thinklab-SJTU/Crossformer
- FEDformer: https://github.com/MAZiqing/FEDformer
- Autoformer: https://github.com/thuml/Autoformer
- Informer: https://github.com/zhouhaoyi/Informer2020

It should be noted that reproducing these experiments requires a significant amount of computational resources. To save computational resources, the data for FEDformer, Autoformer, and Informer were

directly sourced from PatchTST (Nie et al., 2023) since its reproduction strategy is consistent with our work.

### A.1.3    CONFIGURATIONS

By default, PETformer utilizes 4 layers of Transformer encoder with a hidden dimension of 512, multi-head attention with 8 heads, a dropout rate of 0.5, and a feedforward factor of 2. The default LSD strategy employs a patch length of 48. The default loss function is Smooth L1 Loss, with the FA mode in the PET setting and the NCI mode in the MSI setting. For detailed information, please refer to our forthcoming open-source code release.

### A.1.4    ENVIRONMENTS

All experiments in this study are implemented in PyTorch and conducted on two NVIDIA V100 GPUs, each with 16GB of memory.

### A.2    MORE ABLATION STUDIES ABOUT RevIN AND SMOOTH L1 LOSS

We conducted more ablation experiments about RevIN and Smooth L1 loss and compared them with PatchTST, as presented in Table 7. Here are the conclusions drawn:

- Regardless of the use of normalization functions or different loss functions, PETformer outperforms PatchTST.

- Smooth L1 loss combines L1 and L2 losses. Compared to using L2 loss, adopting Smooth L1 loss increases MSE error slightly while decreasing MAE error. Taking a holistic view, we opted for Smooth L1 loss. However, it's important to note that the differences in performance due to different loss functions are generally small, thus the choice of error function isn't the key factor for PETformer's success.

- RevIN is a technique to mitigate distribution shifts in time series, which consistently improves performance across different models. PatchTST defaulted to using RevIN and we followed suit. Therefore, as a technique gaining traction in the LTSF field, RevIN serves as a foundation for PETformer's success, but it is not the key factor. The innovative design of PETformer is what sets it apart from other models.

Table 7: Ablation study of loss functions and RevIN. The best results are highlighted in **bold**.

| Settings | | Smooth L1 Loss + RevIN | | | | L2 Loss + RevIN | | | | Smooth L1 Loss - RevIN | | | |
|---|---|---|---|---|---|---|---|---|---|---|---|---|---|
| Models | | PETformer | | PatchTST | | PETformer | | PatchTST | | PETformer | | PatchTST | |
| Metric | | MSE | MAE | MSE | MAE | MSE | MAE | MSE | MAE | MSE | MAE | MSE | MAE |
| ETTm1 | 96 | **0.282** | **0.325** | 0.292 | 0.339 | **0.280** | **0.337** | 0.301 | 0.353 | **0.287** | **0.338** | 0.311 | 0.363 |
| | 192 | **0.318** | **0.349** | 0.334 | 0.366 | **0.319** | **0.361** | 0.334 | 0.374 | **0.322** | **0.362** | 0.341 | 0.379 |
| | 336 | **0.348** | **0.372** | 0.362 | 0.386 | **0.348** | **0.384** | 0.362 | 0.394 | **0.364** | **0.396** | 0.375 | 0.409 |
| | 720 | **0.404** | **0.403** | 0.422 | 0.416 | **0.405** | **0.416** | 0.421 | 0.422 | **0.412** | **0.416** | 0.421 | 0.42 |
| Traffic | 96 | **0.357** | **0.240** | 0.376 | 0.251 | **0.358** | **0.250** | 0.367 | 0.253 | 0.449 | **0.243** | **0.398** | 0.244 |
| | 192 | **0.376** | **0.248** | 0.390 | 0.255 | **0.374** | **0.255** | 0.382 | 0.259 | 0.470 | **0.250** | **0.414** | 0.252 |
| | 336 | **0.392** | **0.255** | 0.398 | 0.260 | **0.390** | **0.262** | 0.396 | 0.267 | 0.497 | **0.260** | **0.435** | **0.260** |
| | 720 | **0.430** | **0.276** | 0.437 | 0.282 | **0.428** | **0.283** | 0.433 | 0.287 | 0.548 | **0.285** | **0.512** | **0.285** |
| Weather | 96 | **0.146** | **0.186** | 0.148 | 0.191 | **0.146** | **0.192** | 0.147 | 0.198 | **0.144** | **0.192** | 0.154 | 0.207 |
| | 192 | **0.190** | **0.229** | 0.193 | 0.234 | **0.190** | **0.234** | 0.190 | 0.241 | **0.194** | **0.248** | 0.194 | 0.243 |
| | 336 | **0.241** | **0.271** | 0.243 | 0.273 | **0.239** | **0.273** | 0.243 | 0.284 | 0.247 | 0.297 | **0.243** | **0.282** |
| | 720 | **0.314** | **0.323** | 0.311 | 0.326 | **0.308** | **0.324** | 0.305 | 0.328 | **0.306** | **0.337** | 0.309 | 0.341 |

Furthermore, we noted an interesting phenomenon: When RevIN is not used, PETformer's MAE is nearly on par with PatchTST on the Traffic dataset, but there is a significant difference in MSE. We speculate this might be due to the window-wise prediction approach causing more oscillations in edge predictions, leading to higher MSE values. Of course, these differences require further in-depth investigation in future work.

Table 8: The performance of PETformer on different look-back window lengths of $l \in \{48, 96, 192, 336, 480, 720\}$. The best results are highlighted in **bold**.

| Input Length | | 48 | | 96 | | 192 | | 336 | | 480 | | 720 | |
|---|---|---|---|---|---|---|---|---|---|---|---|---|---|
| Metric | | MSE | MAE | MSE | MAE | MSE | MAE | MSE | MAE | MSE | MAE | MSE | MAE |
| ETTh1 | 96 | 0.389 | 0.392 | 0.376 | 0.387 | 0.369 | 0.385 | 0.358 | 0.381 | 0.353 | 0.380 | **0.347** | **0.377** |
| | 192 | 0.443 | 0.424 | 0.429 | 0.417 | 0.414 | 0.410 | 0.397 | 0.404 | **0.388** | **0.402** | 0.390 | 0.404 |
| | 336 | 0.488 | 0.445 | 0.467 | 0.433 | 0.437 | 0.420 | 0.419 | **0.417** | 0.422 | 0.423 | **0.419** | 0.418 |
| | 720 | 0.482 | 0.462 | 0.471 | 0.459 | 0.449 | 0.450 | 0.443 | 0.453 | 0.438 | **0.447** | **0.437** | 0.449 |
| ETTh2 | 96 | 0.299 | 0.340 | 0.288 | 0.339 | 0.284 | 0.338 | 0.281 | 0.333 | **0.270** | **0.329** | 0.272 | 0.329 |
| | 192 | 0.383 | 0.393 | 0.367 | 0.388 | 0.353 | 0.381 | 0.345 | 0.376 | 0.338 | **0.374** | **0.338** | 0.374 |
| | 336 | 0.397 | 0.410 | 0.369 | 0.398 | 0.349 | 0.394 | 0.336 | 0.380 | 0.330 | **0.378** | **0.328** | 0.380 |
| | 720 | 0.426 | 0.438 | 0.409 | 0.433 | 0.391 | 0.423 | **0.385** | **0.422** | 0.387 | 0.426 | 0.401 | 0.439 |
| ETTm1 | 96 | 0.468 | 0.413 | 0.316 | 0.341 | 0.292 | 0.328 | 0.281 | 0.324 | **0.279** | **0.323** | 0.282 | 0.325 |
| | 192 | 0.514 | 0.437 | 0.366 | 0.367 | 0.333 | 0.353 | 0.321 | 0.351 | 0.320 | 0.349 | **0.318** | **0.349** |
| | 336 | 0.554 | 0.462 | 0.398 | 0.390 | 0.366 | 0.376 | 0.356 | 0.372 | 0.352 | **0.370** | 0.348 | 0.372 |
| | 720 | 0.601 | 0.489 | 0.463 | 0.427 | 0.425 | 0.412 | 0.416 | 0.407 | 0.415 | 0.407 | **0.404** | **0.403** |
| ETTm2 | 96 | 0.190 | 0.271 | 0.174 | 0.255 | 0.167 | 0.248 | 0.160 | **0.245** | 0.163 | 0.248 | **0.160** | 0.248 |
| | 192 | 0.258 | 0.313 | 0.241 | 0.298 | 0.230 | 0.289 | 0.220 | 0.289 | 0.221 | **0.288** | **0.217** | 0.288 |
| | 336 | 0.324 | 0.353 | 0.298 | 0.335 | 0.283 | 0.327 | 0.271 | 0.320 | **0.269** | **0.320** | 0.274 | 0.326 |
| | 720 | 0.425 | 0.409 | 0.393 | 0.392 | 0.366 | 0.382 | 0.357 | 0.379 | 0.352 | 0.378 | **0.345** | **0.376** |
| Electricity | 96 | 0.222 | 0.292 | 0.171 | 0.254 | 0.138 | 0.229 | 0.131 | 0.223 | 0.128 | 0.221 | **0.128** | **0.220** |
| | 192 | 0.217 | 0.290 | 0.179 | 0.263 | 0.154 | 0.244 | 0.147 | 0.238 | 0.145 | 0.236 | **0.144** | **0.236** |
| | 336 | 0.235 | 0.307 | 0.195 | 0.279 | 0.177 | 0.268 | 0.163 | 0.255 | 0.161 | 0.253 | **0.159** | **0.252** |
| | 720 | 0.272 | 0.337 | 0.234 | 0.311 | 0.206 | 0.291 | 0.203 | 0.289 | 0.198 | 0.286 | **0.195** | **0.286** |
| Traffic | 96 | 0.671 | 0.379 | 0.465 | 0.299 | 0.396 | 0.260 | 0.373 | 0.249 | 0.365 | 0.245 | **0.357** | **0.240** |
| | 192 | 0.617 | 0.351 | 0.470 | 0.294 | 0.415 | 0.267 | 0.392 | 0.256 | 0.386 | 0.253 | **0.376** | **0.248** |
| | 336 | 0.640 | 0.362 | 0.486 | 0.301 | 0.430 | 0.274 | 0.403 | 0.261 | 0.396 | 0.258 | **0.392** | **0.255** |
| | 720 | 0.668 | 0.371 | 0.518 | 0.318 | 0.457 | 0.289 | 0.436 | 0.279 | 0.432 | 0.278 | **0.430** | **0.276** |
| Weather | 96 | 0.210 | 0.240 | 0.178 | 0.214 | 0.162 | 0.200 | 0.150 | 0.190 | 0.148 | 0.188 | **0.146** | **0.186** |
| | 192 | 0.245 | 0.269 | 0.225 | 0.254 | 0.205 | 0.240 | 0.194 | 0.232 | 0.192 | 0.229 | **0.190** | **0.229** |
| | 336 | 0.298 | 0.307 | 0.278 | 0.294 | 0.258 | 0.280 | 0.246 | 0.273 | 0.243 | **0.271** | **0.241** | 0.271 |
| | 720 | 0.372 | 0.355 | 0.352 | 0.344 | 0.333 | 0.333 | 0.320 | 0.326 | 0.315 | 0.324 | **0.314** | **0.323** |
| **Avg.** | | 0.404 | 0.368 | 0.342 | 0.339 | 0.316 | 0.325 | 0.304 | 0.319 | 0.300 | 0.317 | **0.298** | **0.317** |

## A.3 VARYING LOOK-BACK WINDOW

The length of the look-back window is a crucial factor in time series forecasting tasks, as it determines the amount of past data that the model can incorporate. Generally, a model that can effectively capture long-term temporal patterns is expected to exhibit better performance as the look-back window length increases. Therefore, we investigated the performance of PETformer under varying look-back window lengths, specifically $l \in \{48, 96, 192, 336, 480, 720\}$. The experimental results, as shown in Table 8, indicate that the performance of PETformer continuously improves as the look-back window length increases. This demonstrates PETformer's ability to model long-term dependencies in time-series data effectively.

## A.4 MULTI-CHANNEL SEPARATION

In this sub-section, we investigate why the MSI design can enhance predictive performance. We believe that channel-separation strategy can enlarge the number of training samples and thereby enhance the model's ability to predict individual channels, leading to an overall improvement in the predictive ability across multiple channels.

Table 9 defines Univariate as the univariate prediction based solely on the last channel of the dataset. In contrast, MSI/NCI can be regarded as a pure univariate prediction task since it does not involve any inter-channel interaction. However, it significantly increases the number of training samples for univariate prediction tasks, as it utilizes data from all channels to perform univariate prediction. We can see that MSI/NCI exhibits a considerable improvement over Univariate, demonstrating that simply increasing the number of samples (even if they originate from other dimensions of the same dataset) can enhance the model's ability to predict a single channel. Note that the ETTh1, Weather, Electricity, and Traffic datasets contain 7, 21, 321, and 862 multivariate variables, respectively. Based on Table 9, we can observe that for the ETTh1 dataset, which contains only 7 variables, the MSI design even exhibits a decline in performance compared to Univariate. However, when the

Table 9: Results of PETformer on single-variable prediction under different modes. "Univariate" refers to the conventional univariate prediction task on the last dimension of the multivariate dataset. The remaining modes are trained in a multivariate prediction manner and then predict the last dimension of the multivariate dataset..

| Models | | Univariate | | MSI/NCI | | MSI/CI | | MSI/SA | | MSI/CA | | DCM | |
|---|---|---|---|---|---|---|---|---|---|---|---|---|---|
| Metric | | MSE | MAE | MSE | MAE | MSE | MAE | MSE | MAE | MSE | MAE | MSE | MAE |
| ETTh1 | 96 | 0.052 | 0.174 | 0.051 | **0.170** | **0.050** | 0.171 | 0.051 | 0.171 | 0.051 | 0.170 | 0.053 | 0.176 |
| | 192 | **0.066** | 0.201 | 0.067 | **0.197** | 0.067 | 0.198 | 0.068 | 0.199 | 0.068 | 0.198 | 0.070 | 0.203 |
| | 336 | **0.075** | **0.217** | 0.081 | 0.224 | 0.079 | 0.222 | 0.080 | 0.223 | 0.082 | 0.225 | 0.076 | 0.218 |
| | 720 | **0.079** | **0.225** | 0.092 | 0.238 | 0.084 | 0.228 | 0.093 | 0.240 | 0.094 | 0.240 | 0.115 | 0.266 |
| Weather | 96 | 0.0014 | 0.028 | 0.0008 | 0.020 | 0.0008 | 0.020 | 0.0008 | 0.020 | **0.0008** | **0.019** | 0.0012 | 0.025 |
| | 192 | 0.0013 | 0.026 | 0.0011 | 0.023 | 0.0011 | **0.023** | 0.0011 | 0.023 | **0.0011** | 0.023 | 0.0014 | 0.027 |
| | 336 | 0.0015 | 0.029 | 0.0013 | 0.026 | 0.0013 | 0.026 | 0.0013 | 0.026 | **0.0013** | **0.026** | 0.0018 | 0.031 |
| | 720 | 0.0020 | 0.034 | 0.0018 | 0.031 | 0.0018 | **0.031** | 0.0018 | 0.031 | **0.0018** | 0.031 | 0.0025 | 0.037 |
| Electricity | 96 | 0.213 | 0.312 | 0.185 | 0.293 | **0.183** | **0.292** | 0.185 | 0.292 | 0.187 | 0.292 | 0.680 | 0.642 |
| | 192 | 0.244 | 0.340 | 0.218 | 0.318 | **0.215** | **0.317** | 0.220 | 0.319 | 0.227 | 0.327 | 0.668 | 0.633 |
| | 336 | 0.293 | 0.389 | 0.254 | 0.348 | **0.248** | **0.345** | 0.250 | 0.347 | 0.258 | 0.357 | 0.663 | 0.631 |
| | 720 | 0.491 | 0.512 | 0.299 | 0.403 | 0.290 | 0.396 | 0.280 | 0.388 | **0.265** | **0.372** | 0.628 | 0.622 |
| Traffic | 96 | 0.118 | 0.195 | 0.101 | 0.166 | 0.101 | **0.165** | 0.101 | 0.166 | 0.108 | 0.178 | 0.582 | 0.580 |
| | 192 | 0.119 | 0.195 | **0.106** | **0.170** | 0.106 | 0.172 | 0.107 | 0.175 | 0.115 | 0.185 | 0.611 | 0.596 |
| | 336 | 0.120 | 0.198 | 0.106 | 0.174 | 0.105 | 0.173 | **0.104** | **0.173** | 0.114 | 0.189 | 0.627 | 0.606 |
| | 720 | 0.140 | 0.224 | 0.124 | 0.197 | 0.122 | 0.194 | **0.119** | **0.191** | 0.132 | 0.208 | 0.559 | 0.566 |
| **Avg.** | | 0.126 | 0.206 | 0.106 | 0.187 | **0.104** | **0.186** | 0.104 | 0.187 | 0.107 | 0.190 | 0.334 | 0.366 |

number of variables in the dataset increases significantly, resulting in a corresponding increase in the number of training samples, MSI/NCI can achieve a substantial improvement over Univariate.

MSI/CI represents a further strategy, which adds channel identifiers to each channel on the basis of MSI/NCI. Naturally, this further enhances the predictive ability on individual channels because it provides the model with identification information for different channels.

In addition, MSI/SA, MSI/CA, and DCM can be regarded as multivariate-predict-univariate tasks because they take into the interaction between channels. Here, MSI/SA and MSI/CA are both better than Univariate, which indicates the effectiveness of the MSI design. However, DCM performs relatively poorly, with much worse performance than Univariate. This fully demonstrates that the direct mixing of channels can seriously disrupt the temporal information within individual channels.

## A.5 ROBUSTNESS ANALYSIS

To ensure the robustness of our experimental findings, we conducted each experiment three times using different random seeds: 2023, 2024, and 2025. The mean and standard deviation of the results are summarized in Table 10, which shows that the variances are notably small. This indicates that our model is robust to the choice of random seeds, and the reported results can be considered reliable. Both the main text and the appendix present results obtained using a fixed random seed of 2023, and these results are consistent with the results obtained using the other random seeds.

## A.6 FORECAST SHOWCASES

In this sub-section, we present the prediction showcases of PETformer on the large datasets, as shown in Figure 5 and Figure 6. We will illustrate this using the Electricity dataset as an example, where PETformer successfully captures several crucial features:

- Periodicity: Electricity consumption exhibits daily cycles, with consumption rising during the daytime, slightly decreasing around noon, and significantly dropping at night, except for a slight increase in the evening. These patterns align with actual electricity consumption trends, and PETformer successfully captures this periodicity.

- Seasonality: Electricity consumption also displays seasonal variations, such as higher usage on weekdays compared to weekends. PETformer's predictions accurately capture these variations.

Table 10: Quantitative results of PETformer with fluctuations across different random seeds: 2023, 2024, and 2025.

| Random seed | | 2023 | | 2024 | | 2025 | | Standard deviation | |
|---|---|---|---|---|---|---|---|---|---|
| Metric | | MSE | MAE | MSE | MAE | MSE | MAE | MSE | MAE |
| ETTh1 | 96 | 0.347 | 0.377 | 0.348 | 0.379 | 0.350 | 0.379 | 0.348±0.0014 | 0.378±0.0010 |
| | 192 | 0.390 | 0.404 | 0.388 | 0.402 | 0.392 | 0.407 | 0.390±0.0020 | 0.404±0.0021 |
| | 336 | 0.419 | 0.418 | 0.418 | 0.421 | 0.414 | 0.416 | 0.417±0.0028 | 0.419±0.0027 |
| | 720 | 0.437 | 0.449 | 0.474 | 0.473 | 0.436 | 0.448 | 0.449±0.0216 | 0.457±0.0139 |
| ETTh2 | 96 | 0.272 | 0.329 | 0.270 | 0.329 | 0.272 | 0.330 | 0.271±0.0011 | 0.329±0.0006 |
| | 192 | 0.338 | 0.374 | 0.338 | 0.375 | 0.334 | 0.372 | 0.336±0.0019 | 0.374±0.0014 |
| | 336 | 0.328 | 0.380 | 0.326 | 0.379 | 0.331 | 0.384 | 0.328±0.0025 | 0.381±0.0026 |
| | 720 | 0.401 | 0.439 | 0.399 | 0.437 | 0.398 | 0.436 | 0.399±0.0016 | 0.437±0.0019 |
| ETTm1 | 96 | 0.282 | 0.325 | 0.281 | 0.324 | 0.277 | 0.324 | 0.280±0.0023 | 0.324±0.0007 |
| | 192 | 0.318 | 0.349 | 0.317 | 0.349 | 0.318 | 0.349 | 0.318±0.0007 | 0.349±0.0005 |
| | 336 | 0.348 | 0.372 | 0.347 | 0.370 | 0.349 | 0.370 | 0.348±0.0010 | 0.371±0.0009 |
| | 720 | 0.404 | 0.403 | 0.405 | 0.402 | 0.405 | 0.402 | 0.405±0.0005 | 0.402±0.0001 |
| ETTm2 | 96 | 0.160 | 0.248 | 0.161 | 0.246 | 0.159 | 0.246 | 0.160±0.0007 | 0.247±0.0008 |
| | 192 | 0.217 | 0.288 | 0.219 | 0.290 | 0.218 | 0.287 | 0.218±0.0009 | 0.288±0.0016 |
| | 336 | 0.274 | 0.326 | 0.279 | 0.328 | 0.276 | 0.326 | 0.276±0.0024 | 0.327±0.0015 |
| | 720 | 0.345 | 0.376 | 0.345 | 0.377 | 0.352 | 0.380 | 0.348±0.0041 | 0.378±0.0022 |
| Electricity | 96 | 0.128 | 0.220 | 0.128 | 0.220 | 0.128 | 0.220 | 0.128±0.0002 | 0.220±0.0002 |
| | 192 | 0.144 | 0.236 | 0.144 | 0.236 | 0.144 | 0.236 | 0.144±0.0002 | 0.236±0.0002 |
| | 336 | 0.159 | 0.252 | 0.158 | 0.252 | 0.158 | 0.251 | 0.159±0.0004 | 0.252±0.0006 |
| | 720 | 0.195 | 0.286 | 0.193 | 0.284 | 0.194 | 0.284 | 0.194±0.0008 | 0.285±0.0010 |
| Traffic | 96 | 0.357 | 0.240 | 0.359 | 0.241 | 0.359 | 0.241 | 0.358±0.0010 | 0.241±0.0005 |
| | 192 | 0.376 | 0.248 | 0.377 | 0.249 | 0.376 | 0.248 | 0.376±0.0006 | 0.249±0.0003 |
| | 336 | 0.392 | 0.255 | 0.391 | 0.255 | 0.391 | 0.255 | 0.391±0.0004 | 0.255±0.0002 |
| | 720 | 0.430 | 0.276 | 0.431 | 0.277 | 0.429 | 0.276 | 0.430±0.0006 | 0.276±0.0004 |
| Weather | 96 | 0.146 | 0.186 | 0.145 | 0.184 | 0.146 | 0.184 | 0.145±0.0002 | 0.185±0.0009 |
| | 192 | 0.190 | 0.229 | 0.189 | 0.228 | 0.190 | 0.228 | 0.190±0.0007 | 0.228±0.0005 |
| | 336 | 0.241 | 0.271 | 0.243 | 0.272 | 0.242 | 0.271 | 0.242±0.0009 | 0.271±0.0005 |
| | 720 | 0.314 | 0.323 | 0.315 | 0.324 | 0.315 | 0.324 | 0.315±0.0004 | 0.324±0.0004 |

- Trends: Overall trends in electricity consumption are also present. In the Horizon-720 graph, the user's consumption trend remains stable, and PETformer's predictions align with this trend.

These results underscore PETformer's ability to effectively learn and model the underlying trends and periodic patterns within time series data, thereby enabling accurate predictions of future data trends.

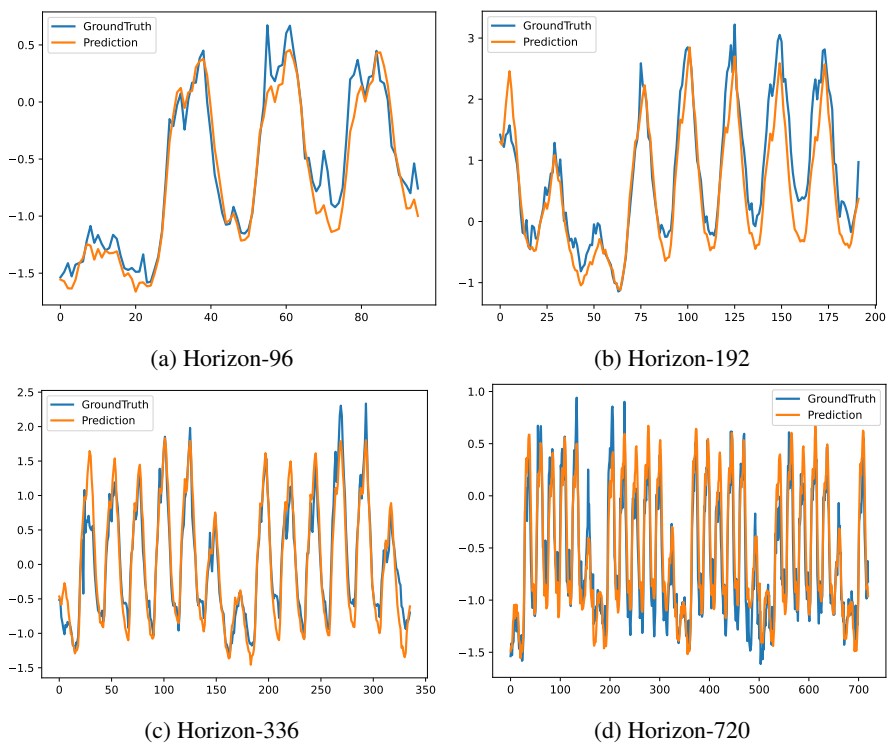

Figure 5: Prediction cases for the Electricity dataset with a look-back window length of $l = 720$ and a forecast horizon of $h \in \{96, 192, 336, 720\}$.

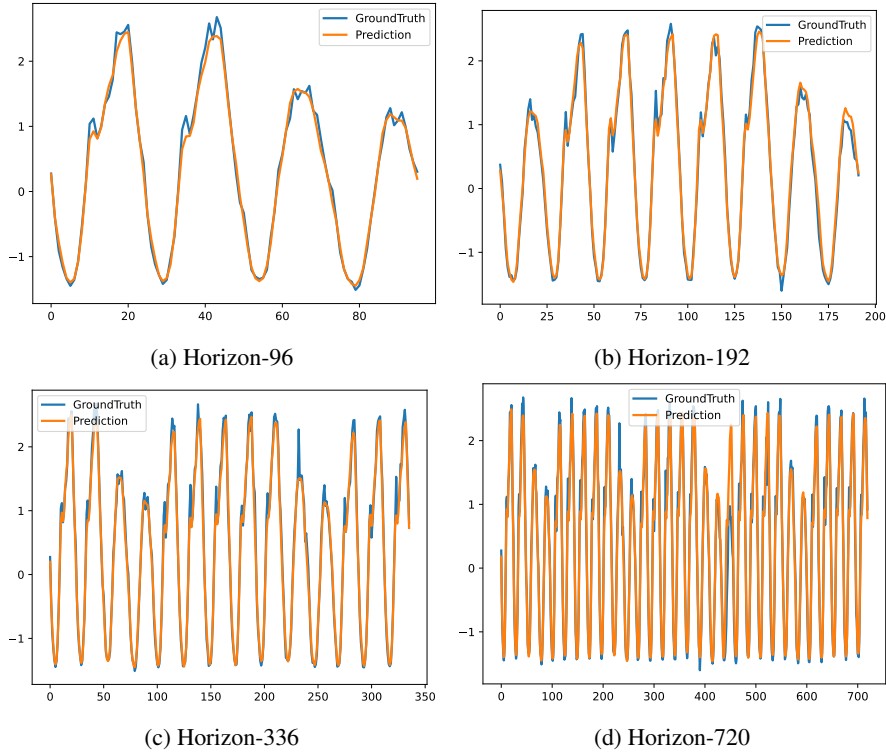

Figure 6: Prediction cases for the Traffic dataset with a look-back window length of $l = 720$ and a forecast horizon of $h \in \{96, 192, 336, 720\}$.

