# OpenReview forum: "PETformer: Long-term Time Series Forecasting via Placeholder-enhanced Transformer"
_ICLR.cc/2024/Conference — ICLR 2024 Conference Withdrawn Submission_

### Official Review · Reviewer_27hd · 2023-10-19

**Soundness:** 2 fair
**Presentation:** 1 poor
**Contribution:** 1 poor
**Rating:** 3
**Confidence:** 4

**Summary:**

This paper proposes a novel Transformer-based model for long-term time series forecasting (LTSF), which introduce the Placeholder-enhanced Technique (PET) to enhance the computational efficiency and predictive accuracy, and delves into different strategies related to Transformer. Experiments on multiple real-world datasets have demonstrated that PETformer achieves good performance for LTSF.

**Strengths:**

1.The idea of using “Placeholder” technique in LTSF is novel.

2.The experimental performance of the proposed model beats most of the baselines and shows promising results on several datasets. The ablation study seems detailed and comprehensive.

**Weaknesses:**

1.The purpose of this paper is unclear: what problems are this paper trying to solve? To prove the effectiveness of Transformer in LSTF, or to enhance the model PatchTST? The unclear purpose causes confusions in the introduction section (for example, the introduction mentions the model Dlinear, but what is the relationship between PETformer and Dlinear)?

2.The paper does not sufficiently analyze why the "Placeholder" technique improves the model's performance. The only sentence I find is “Allowing more learnable parameters to influence the feature extraction module of the Transformer, rather than the linear prediction head, may be a key reason for PETformer outperforming PatchTST”, but that’s not enough. The reason why the “Placeholder” technique is effective should be the core of the paper.

3.The discrepancy between the results in Table 2 and Table 4 needs to be addressed with an explanation.

4.The paper should provide information on the initial values in the "Placeholder," and whether different values affect the model's performance.

5.The experiments used the look-back window size of 720 in most datasets and claimed to achieve SOTA performance. However, this comparison may be unfair as a longer look-back window generally provides more information and potentially better results. Additionally, it is essential to investigate if PatchTST performs better with a look-back window size of 720.

6.The paper mentions that “The Placeholder technique shares similarities with the currently popular Masking technique in the unsupervised pretraining domain”, however, this claim requires further explanation to strengthen its validity.

7.Some figures in the paper require enhancement. For example, the left sub-figure of Figure 2 should include the ‘Placeholder’ in the input, and the middle sub-figure of Figure 2 should display the ‘Placeholder’ with the same length as the output but with a different color.
The details of the ablation studies need to be clearly described, and there are too many abbreviations of self-defined terms (NIFA, NIHA, OFFH, SA, CA ...) in the paper, which may inconvenience readers.

**Questions:**

Please refer to the Weaknesses.

---

### Official Review · Reviewer_Gtuh · 2023-10-27

**Soundness:** 3 good
**Presentation:** 3 good
**Contribution:** 2 fair
**Rating:** 5
**Confidence:** 5

**Summary:**

The paper proposes a Transformer-based model (PETformer) for Long-Term Time Series Forecasting. PETformer concatenates several learnable placeholders of future sequences and embeddings of past sequences, and inputs them into the Transformer encoder. This approach maintains the temporal continuity between past and predicted sequences, and significantly reduces the number of parameters required for the prediction head. Additionally, this paper explores the impact of multi-channel relationships and long subsequences on time series prediction tasks. Experimental validation on eight public datasets confirms the effectiveness of PETformer.

**Strengths:**

1.	The overall writing is good. The logic is clear and easy to follow.
2.	The proposed model, PETformer, achieves the SOTA results on eight public datasets. The authors compared with many advanced prediction models and reached consistent conclusions.
3.	The ablation study is very comprehensive, which proves the rationality and effectiveness of the model design.

**Weaknesses:**

1.	My biggest concern is the novelty of the paper. Compared to PatchTST, the main difference of PETformer is the removal of the flattened prediction head and the use of placeholder embeddings for prediction. As the authors mentioned, this idea has been explored in many other fields, such as Masked Language Modeling in NLP and Masked Image Modeling in CV. It is not uncommon to mask future tokens for prediction, as mentioned in [1]. Additionally, although the authors also explored Long Sub-sequence Division (LSD) and Multi-channel Separation and Interaction (MSI), these are not new in time series forecasting and are only experimental supplements in my view.
2.	When exploring the relationships between multiple channels, the authors only studied the relationships in the predicted sequence, without directly studying the past sequence. In fact, the interaction between multiple channels in the past sequence may also be helpful for prediction. Besides, there is another way of extracting inter-channel features that the authors did not consider. Please refer to Crossformer [2]. Therefore, these omissions may not support the argument about Multi-channel Separation and Interaction (MSI) in the paper.
3.	While the paper presents a comprehensive evaluation of the proposed approach, it does not discuss the limitations or potential drawbacks of the approach.
[1] Gupta A, Tian S, Zhang Y, et al. MaskViT: Masked Visual Pre-Training for Video Prediction. ICLR 2023.
[2] Zhang Y, Yan J. Crossformer: Transformer utilizing cross-dimension dependency for multivariate time series forecasting. ICLR 2023.

**Questions:**

1.	Although predicting with placeholders greatly reduces the number of parameters in the prediction head, it increases the number of tokens involved in self-attention. As we all know, the efficiency bottleneck of Transformer is self-attention. Therefore, my question is why the computational efficiency still improves after increasing the number of tokens?
2.	In the study of the window size w of the sub-sequence, if the length is fixed at 720, will the performance of LTSF continue to increase as w continues to increase?

---

### Official Review · Reviewer_vDyH · 2023-11-04

**Soundness:** 3 good
**Presentation:** 2 fair
**Contribution:** 2 fair
**Rating:** 5
**Confidence:** 4

**Summary:**

This paper proposes Placeholder-enhanced Technique to enhance the computational efficiency and predictive accuracy of Transformer in LTSF tasks. This paper also studies the impact of larger patch strategies and channel interaction strategies on Transformer’s performance. Experimental results showed the efficacy of proposed method and state-of-the-art performance.

**Strengths:**

1) It's novel to propose Placeholder-enhanced Technique to enhance the computational efficiency and predictive accuracy of Transformer to time series forecasting.

2) Experimental results showed state-of-the-art performance.

**Weaknesses:**

The current time series forecasting datasets are pretty small, and performance may be satuated or over-fitting.

**Questions:**

The current time series forecasting datasets are pretty small, and performance may be satuated or over-fitting.  Could this method be used for larger datasets?